# High-Density Lipoprotein Changes in Alzheimer’s Disease Are *APOE* Genotype-Specific

**DOI:** 10.3390/biomedicines10071495

**Published:** 2022-06-24

**Authors:** Brian V. Hong, Jingyuan Zheng, Joanne K. Agus, Xinyu Tang, Carlito B. Lebrilla, Lee-Way Jin, Izumi Maezawa, Kelsey Erickson, Danielle J. Harvey, Charles S. DeCarli, Dan M. Mungas, John M. Olichney, Sarah T. Farias, Angela M. Zivkovic

**Affiliations:** 1Department of Nutrition, University of California-Davis, Davis, CA 95616, USA; bvhong@ucdavis.edu (B.V.H.); jaczheng@ucdavis.edu (J.Z.); jkagus@ucdavis.edu (J.K.A.); xctang@ucdavis.edu (X.T.); 2Department of Chemistry, University of California-Davis, Davis, CA 95616, USA; cblebrilla@ucdavis.edu; 3Department of Pathology and Laboratory Medicine, School of Medicine, University of California-Davis, Davis, CA 95817, USA; lwjin@ucdavis.edu (L.-W.J.); imaezawa@ucdavis.edu (I.M.); kelerickson@ucdavis.edu (K.E.); 4Department of Public Health Sciences, University of California-Davis, Davis, CA 95616, USA; djharvey@ucdavis.edu; 5Department of Neurology, School of Medicine, University of California-Davis, Davis, CA 95817, USA; cdecarli@ucdavis.edu (C.S.D.); dmmungas@ucdavis.edu (D.M.M.); jmolichney@ucdavis.edu (J.M.O.); farias@ucdavis.edu (S.T.F.)

**Keywords:** Alzheimer’s disease, APOE, cholesterol efflux capacity, HDL, LCAT

## Abstract

High-density lipoproteins (HDL) play a critical role in cholesterol homeostasis. Apolipoprotein E (*APOE)*, particularly the *E4* allele, is a significant risk factor for Alzheimer’s disease but is also a key HDL-associated protein involved in lipid transport in both the periphery and central nervous systems. The objective was to determine the influence of the *APOE* genotype on HDL function and size in the context of Alzheimer’s disease. HDL from 194 participants (non-demented controls, mild cognitive impairment, and Alzheimer’s disease dementia) were isolated from the plasma. The HDL cholesterol efflux capacity (CEC), lecithin-cholesterol acyltransferase (LCAT) activity, and particle diameter were measured. Neuropsychological test scores, clinical dementia rating, and magnetic resonance imaging scores were used to determine if cognition is associated with HDL function and size. HDL CEC and LCAT activity were reduced in *APOE3E4* carriers compared to *APOE3E3* carriers, regardless of diagnosis. In *APOE3E3* carriers, CEC and LCAT activity were lower in patients. In *APOE3E4* patients, the average particle size was lower. HDL LCAT activity and particle size were positively correlated with the neuropsychological scores and negatively correlated with the clinical dementia rating. We provide evidence for the first time of *APOE* genotype-specific alterations in HDL particles in Alzheimer’s disease and an association between HDL function, size, and cognitive function.

## 1. Introduction

Alzheimer’s disease is the leading cause of dementia and poses a considerable economic and public health burden [1]. Alzheimer’s disease develops over the course of decades [2], and drugs targeting late-stage processes once dementia and brain volume loss have set in may be “too late”. There is an urgent need for effective, long-term treatments that target the underlying pathophysiology of Alzheimer’s disease to prevent cognitive function loss and the onset of dementia.

Increasing evidence suggests that high-density lipoprotein (HDL) particles, both in the central nervous system (CNS) and in the periphery, are implicated in Alzheimer’s disease pathology. Alzheimer’s disease patients typically have lower plasma HDL cholesterol (HDL-C) and apolipoprotein-A1 (ApoA-I) concentrations than controls [3,4], suggesting that a lack of HDL is detrimental. HDL-C concentrations measured during midlife are negatively correlated with the onset of late-life mild cognitive impairment (MCI) and dementia [5], suggesting that HDL are protective. However, the functional capacity of HDL is a stronger predictor of disease risk than HDL-C concentrations, since the measurement of HDL-C does not encompass the complex, destructive changes to HDL particles that can occur in disease states and in aging [6,7]. The main function of HDL particles is to mediate reverse cholesterol transport, though they perform numerous additional protective functions, including reducing inflammation, promoting endothelial function, and antioxidant effects, among others [8].

Although ApoA-I is not expressed in the CNS, the peripheral overexpression of human ApoA-I preserves cognitive function, reduces neuroinflammation, and protects mice from cerebral amyloid angiopathy [9], suggesting a role for peripheral HDL in the clearance of brain amyloid beta (Aβ). In a small study of 39 Alzheimer’s disease patients and 20 healthy participants (unknown *APOE* genotype), HDL isolated from plasma showed a diminished HDL cholesterol efflux capacity (CEC) and lecithin-cholesterol acyltransferase (LCAT) activity—the enzyme responsible for increasing the HDL cholesterol carrying capacity—and the level of LCAT activity was negatively correlated with the cognitive score [10]. Together, these studies suggest that circulating HDL is involved in the pathology of Alzheimer’s disease. However, the relationship between the peripheral HDL functional capacity and Alzheimer’s disease has not been fully investigated in a larger cohort of Alzheimer’s disease patients, and the effects of the *APOE* genotype on this relationship have not been investigated.

In the current study, we determined whether HDL CEC and LCAT activity are altered in a large *APOE* genotyped cohort of elderly participants clinically diagnosed as either non-demented, MCI, or Alzheimer’s disease dementia (AD). We also determined whether the HDL particle size, which is closely related to the HDL function, was altered in the same cohort. We further explored whether the HDL functional metrics and particle size were associated with the participants’ cognitive, functional, and imaging scores.

## 2. Materials and Methods

### 2.1. Participants

This study used plasma samples collected from the University of California, Davis Alzheimer’s Disease Research Center (ADRC) Biorepository. The ADRC biorepository is nationally recognized for recruiting an ethnically diverse clinic-based and community-based elderly cohort [11]. We aimed to select 200 samples from the ADRC biorepository to include non-demented (controls), MCI, and AD patients, with each diagnosis group having as close as possible to equal numbers of participants with the *APOE3E3* and *APOE3E4* genotypes and with each genotype X diagnosis group having as close as possible to equal numbers of the sexes (males and females), with equal average ages. Only participants with an adequate sample volume available (500 μL) and body mass index (BMI) not greater than 40 kg/m^2^ were included. Participants were diagnosed following the ADRC criteria for a clinical diagnosis within one year of a blood draw, as previously described [12]. Dementia patients were included in the AD group if their etiologic diagnosis was classified as “probable” or “possible” AD. Five patients in the AD group had an etiologic diagnosis of frontotemporal dementia. Removing these patients did not change the overall findings; thus, they were included in the analysis. Samples from a total of 194 participants met the above criteria and were included in the final analysis. The study was approved by the Institutional Review Board of the University of California, Davis.

### 2.2. HDL Isolation and Analysis

#### 2.2.1. Isolation Method

HDL particles from plasma were isolated by two-step sequential flotation density-ultracentrifugation, followed by size exclusion chromatography, to yield highly purified HDL fractions, as previously described [13]. Following the manufacturer’s instructions, the total HDL protein was measured using a Micro BCA protein assay kit (Thermo Fisher Scientific, Waltham, MA, USA, catalog 23235).

#### 2.2.2. Cholesterol Efflux Capacity

HDL cholesterol efflux was measured using a commercially available kit (Abcam, Cambridge, UK, catalog ab19685) with modifications, as previously described [14]. J774A.1 (ATCC, Manassas, VA, USA, catalog TIB-67) were seeded at 100,000 cells in 96-well plates for 4 h in Dulbecco’s Modified Eagle’s Medium containing 10% fetal bovine serum and 100 μg/mL penicillin and streptomycin. The cells were washed and labeled with fluorescent cholesterol, acyl-coenzyme A:cholesterol acyltransferase inhibitor, and cyclic adenosine monophosphate for 4 h, followed by wash and incubation with 10 μg of HDL protein for another 4 h. The cellular supernatant was removed, and the remaining cells were lysed with M-PER cell lysis buffer (Thermo Fisher Scientific, Waltham, MA, USA, catalog 78505). The supernatant and lysed fraction fluorescence were measured at 482/515 nm (emission/excitation) on a Synergy H1 plate reader (BioTek, Winooski, VT, USA). To account for the inter-plate variability, CEC values were normalized, with the CEC value of HDL isolated from pooled plasma collected from healthy volunteers on each plate to calculate a final CEC Index, as previously described [15].

#### 2.2.3. LCAT Activity

A commercially available kit was used to measure the LCAT activity (Roar Biomedical, Millipore Sigma, Burlington, MA, USA, catalog mak107) in 5 μg of HDL (measured by the total protein) following the manufacturer’s instructions. Measurements were collected on a Synergy H1 plate reader (BioTek, Winooski, VT, USA) and read at 340-nm excitation and two emission wavelengths at 390 nm and 470 nm, representing the hydrolyzed and intact substrate, respectively. Higher LCAT activity is indicated as increased λ_em_390/λ_em_470 nm ratios.

#### 2.2.4. HDL Particle Size

The HDL particle size was assessed using negative-stained transmission electron microscopy (TEM) based on the published methods [16], in which over 3000 particles were imaged and sized from each individual participant sample. A schematic workflow of the particle size measurements by TEM is presented in Appendix A. Briefly, isolated HDL samples were diluted to 30–100 μg/mL HDL protein concentrations with deionized water. A diluted HDL sample (4 µL) was loaded onto glow-discharged carbon-coated grids (TedPella Inc., Redding, CA, USA) and left for sample attachment for 1 min. Extra sample was removed using filter paper. Uranyl formate solution (2%, pH 7.4) was then added onto the grid and removed 5 times to achieve negative staining. Negatively stained sample grids were then air-dried for 5 min and loaded onto a specimen holder for TEM, according to the manufacturer’s user manual (JEOL USA 1230 Transmission Electron Microscope, JEOL USA Inc., Peabody, MA, USA). Samples were viewed under the TEM using high tension = 100 kv and 40,000× magnification. Micrograph images were taken at random locations using an attached CCD camera (model) with an exposure time of 300 ms.

The software ImageJ [17] was used to characterize the HDL particle size from the TEM micrographs, following a previously published procedure [13], with minor modifications: Noise in the micrographs was first removed using the “Bandpass Filter” function, with “filter large = 100, filter small = 10, suppress = None, and tolerance = 5 autoscale saturate” parameters. The contrast of the cleaned micrograph was then set to “min = 50, max = 205”. The threshold of the micrographs was then set using a premade “Intermodes dark” option. The particle area was then analyzed using the “Analyze Particles” function with “size = 20–7850, circularity = 0.30–1.00, display, exclude, include, and add” parameters. Particles that were captured by the function were then outlined onto the original TEM micrograph and were checked manually for accuracy.

### 2.3. Cognitive Function Analysis

The Spanish English Neuropsychological Assessment Scales (SENAS) evaluated participants’ neuropsychological functional assessments, as previously described and validated elsewhere [18,19]. This study uses the SENAS cognitive domains: verbal memory, executive function, spatial, and semantic memory.

The Clinical Dementia Rating Scale (CDR) is a semi-structured interview administered by a clinician as a global measure of independent function [20]. Six cognitive domains in memory, orientation, judgment and problem-solving, community affairs, home and hobbies, and personal care were assessed. The combination of the scores obtained (“sum of boxes”) was used for analysis.

The white matter hyperintensities (WMH) measurements were acquired by magnetic resonance imaging (MRI), as previously described [21,22]. The total cranial volume was used to normalize the head size differences among participants.

### 2.4. Statistical Analysis

Data analyses were conducted using statistical software R version 4.1.1. (R Project for Statistical Computing, Vienna, Austria). Differences in the HDL functional metrics and particle sizes between AD patients, MCI patients, and controls were tested with one-way ANOVA. The null hypothesis of no difference between groups was rejected if *p* < 0.05. A post hoc Tukey’s HSD was conducted for pairwise comparisons between groups when a significant difference was observed. For two group comparisons, a two-sample *t*-test assuming equal variance was used. For all pairwise comparisons, significance values were reported with *p* < 0.05. Statistically significant findings are indicated as shown: * *p* < 0.05, ** *p* < 0.01, *** *p* < 0.001, and **** *p* < 0.0001. The distribution of the outcome variables and key baseline variables was inspected for normality using the Shapiro–Wilks test. The homogeneity of the variance was examined using Levene’s test. Kruskal–Wallis tests were performed on non-normally distributed data between more than two groups. A chi-squared (χ^2^) test was used to compare the categorical variables.

To examine the association between HDL functional metrics and particle size with cognitive, functional, or MRI assessments, a correlation analysis, adjusting for the *APOE* genotype, was conducted across all diagnoses using Spearman’s correlation. For Spearman’s correlation, the rho (r) values and 95% confidence intervals were reported.

We performed complete case analyses to describe the participant characteristics, HDL functional metrics, and particle size and to assess the association between HDL function and size with cognitive, functional, and MRI assessments. For the HDL particle size analysis, one out of the total of 194 samples was removed due to an insufficient number of total particles on the TEM slide at the time of analysis.

## 3. Results

### 3.1. Participant Characteristics

Details of the participant characteristics and clinical parameters are summarized in Table 1. There were no significant differences observed in the ratio of male to female (χ^2^(2) = 4.10, *p* = 0.129), proportion of ethnicity (χ^2^(6) = 10.97, *p* = 0.089), body mass index (BMI) (*F*(2, 181) = 2.00, *p* = 0.138), or higher history of hypertension (χ^2^(2) = 1.56, *p* = 0.459) among the diagnoses, and these characteristics were not significantly different between groups when stratified by the *APOE* genotype. One-way ANOVA revealed a significant difference in age (*F*(2, 191) = 4.82, *p* = 0.009), with significantly younger participants in the control group than the AD group (mean ± SD age, 75.5 ± 7.0 y vs. 78.9 ± 7.2 y, *p* = 0.008) but not significantly younger than the MCI group (78.0 ± 7.0 y, *p* = 0.153). When the groups were stratified by the *APOE* genotype, one-way ANOVA revealed that none of the age differences among the diagnoses were significantly different within the *APOE3E3* group (*F*(2, 100) = 2.55, *p* = 0.080) or the *APOE3E4* group (*F*(2, 88) = 2.28, *p* = 0.108). There was a significant difference in the prevalence of diabetes at the sample collection (χ^2^(2) = 22.94, *p* < 0.001) and history of diabetes (χ^2^(2) = 28.50, *p* < 0.001) among the groups, with both of these characteristics being higher in the control participants (34% and 39%, respectively) compared to MCI patients (7.9% and 10%, respectively) and AD patients (4.9% and 5.6%, respectively). When stratified by the *APOE* genotype, the prevalence of diabetes at the sample collection was significantly different among the *APOE3E3* carriers (χ^2^(2) = 19.30, *p* < 0.001) but not in the *APOE3E4* carriers (χ^2^(2) = 5.64, *p* = 0.060). Post hoc comparisons in the *APOE3E3* carriers revealed that diabetes during sample collection was higher in the controls (39%) compared with MCI (9.5%) and AD (0%) patients. A history of hypercholesterolemia was significantly different among the diagnoses (χ^2^(2) = 6.70, *p* = 0.035), with a higher history of hypercholesterolemia in the controls compared with AD (74% vs. 54%). When stratified by the *APOE* genotype, a history of hypercholesterolemia was significantly different among the *APOE3E4* carriers (χ^2^(2) = 8.52, *p* = 0.014) but not in the *APOE3E3* carriers (χ^2^(2) = 2.21, *p* = 0.331). A post hoc comparison revealed that the *APOE3E4* controls had a significantly higher history of hypercholesterolemia (87%) than both *APOE3E4* MCI (56%) and AD (62%) patients.

### 3.2. HDL CEC Index and LCAT Activity Differences

The mean ± SD CEC index and LCAT activity are shown in Table 2. When participants were not stratified by the *APOE* genotype, there was no significant difference in the HDL CEC index between the control, MCI, and AD participants by one-way ANOVA (1.10 ± 0.16, 1.06 ± 0.14, and 1.12 ± 0.14, *F*(2, 191) = 2.14, *p* = 0.120, Figure 1A). A one-way ANOVA revealed that there was a significant difference in HDL LCAT activity between at least two groups (*F*(2, 191) = 3.87, *p* = 0.023). Post hoc comparisons indicated the HDL LCAT activity was significantly higher in the control group vs. MCI (1.05 ± 0.09 vs. 1.01 ± 0.07, *p* = 0.030) but not significantly higher than the AD group (1.03 ± 0.07, *p* = 0.118, Figure 1B). There were no significant differences in LCAT activity between MCI and AD (*p* = 0.661). The CEC index (1.06 ± 0.16 vs. 1.13 ± 0.14, *t*(192) = 3.00, *p* = 0.003) and LCAT activity (1.01 ± 0.07 vs. 1.05 ± 0.08, *t*(192) = 3.53, *p* < 0.001) were significantly lower in the *APOE3E4* carriers relative to the *APOE3E3* carriers (Figure 1C,D).

When the participants were stratified by the *APOE* genotype, one-way ANOVA showed a significant difference in the HDL CEC index between at least two groups within the *APOE3E3* carriers (*F*(2, 100) = 3.26, *p* = 0.042) and *APOE3E4* carriers (*F*(2, 88) = 4.38, *p* = 0.015). In the *APOE3E3* group, post hoc comparisons showed the HDL CEC index was lower in MCI patients relative to the controls (1.07 ± 0.13 vs. 1.16 ± 0.13, *p* = 0.042, Figure 2A) but not in AD patients (1.11 ± 0.15) vs. the controls (*p* = 0.246). There was no significant difference in the HDL CEC index between MCI and AD patients (*p* = 0.551). For the *APOE3E4* carriers, AD patients displayed a higher HDL CEC index relative to the controls (1.12 ± 0.14 vs. 1.02 ± 0.16, *p* = 0.016, Figure 2B) and a trend toward a higher efflux in AD vs. MCI patients (1.12 ± 0.14 vs. 1.03 ± 0.14, *p* = 0.114). There was no significant difference in HDL CEC index between the controls and MCI patients (*p* = 0.964).

In the *APOE3E3* carriers, there was a significant difference in HDL LCAT activity between at least two groups by one-way ANOVA (*F*(2, 100) = 7.00, *p* = 0.001). Post hoc comparisons indicate the HDL LCAT activity was reduced in MCI patients relative to the controls (1.03 ± 0.08 vs. 1.09 ± 0.09, *p* = 0.012, Figure 2C) and reduced in AD patients vs. the controls (1.03 ± 0.07 vs. 1.09 ± 0.09, *p* = 0.004), with no difference between MCI and AD patients (*p* = 0.995). When we compared across diagnosis groups within the *APOE3E4* carriers, there was no significant difference in LCAT activity by one-way ANOVA (*F*(2, 88) = 0.91, *p* = 0.410, Figure 2D).

The HDL CEC index and LCAT activity were positively correlated in the *APOE3E3* carriers (r = 0.43, 95% CI (0.26, 0.58), *p* < 0.001, Appendix A), whereas no correlation was observed in the *APOE3E4* carriers (r = 0.16, 95% CI (−0.04, 0.37), *p* = 0.136). When participants were analyzed together regardless of diagnosis or *APOE* genotype, the HDL LCAT activity and CEC index were positively correlated (r = 0.34, 95% CI (0.21, 0.47), *p* < 0.001, Appendix A) and remained correlated after adjusting for the *APOE* genotype (r = 0.31, 95% CI (0.17, 0.42), *p* < 0.001).

### 3.3. HDL Particle Size Differences

The mean ± SD HDL particle size is shown in Table 2. Without stratification for the *APOE* genotype, one-way ANOVA revealed a significant difference in at least two groups (*F*(2, 190) = 9.70, *p* <0.001). Post hoc comparisons showed that the HDL particle sizes in the control group were significantly larger than the MCI (9.06 ± 0.69 nm vs. 8.67 ± 0.63 nm, *p* = 0.008) and the AD group (8.61 ± 0.68 nm, *p* < 0.001, Figure 3A). When stratified by the *APOE* genotype, there were no significant differences among the control, MCI, and AD groups in the *APOE3E3* carriers by one-way ANOVA (8.99 ± 0.85 nm, 8.79 ± 0.58 nm, and 8.72 ± 0.64 nm, respectively, *F*(2, 99) = 1.40, *p* = 0.252, Figure 3B). In the *APOE3E4* carriers, there was a significant difference in HDL particle sizes between at least two groups (*F*(2, 88) = 13.04, *p* < 0.001). Post hoc comparisons indicate that the HDL particle size in the control group was larger than the MCI (9.15 ± 0.42 nm vs. 8.53 ± 0.67 nm, *p* = 0.001) and the AD group (8.50 ± 0.70 nm, *p* < 0.001, Figure 3C). There was no significant difference in the HDL particle size between *APOE3E3* and *APOE3E4* carriers when the diagnosis was not taken into account (8.85 ± 0.73 nm vs. 8.78 ± 0.66 nm, *t*(191) = 0.64, *p* = 0.520).

### 3.4. Correlation between HDL Function, Size, and Cognitive Measures

Details of the participants’ cognitive, functional, and imaging scores are summarized in Table 3. As expected, and by definition, the cognitive, functional, and imaging scores, were significantly differently among the diagnosis groups. The LCAT activity was positively associated with the verbal memory score (r = 0.17, 95% CI (0.01, 0.33), *p* = 0.037) and negatively associated with the CDR (r = −0.20, 95% CI (−0.37, −0.03), *p* = 0.019) and both the verbal memory score (r = 0.18, 95% CI (0.01, 0.33), *p* = 0.033) and CDR (r = −0.20, 95% CI (−0.35, −0.02), *p* = 0.025) remained statistically significant after adjusting for the *APOE* genotype (Table 4). None of the measured cognitive, functional, and imaging scores were significantly correlated with the CEC index. The particle size was positively correlated with the verbal memory score (r = 0.31, 95% CI (0.15, 0.45), *p* < 0.001) and executive function score (r = 0.21, 95% CI (0.04, 0.36), *p* = 0.013) and was negatively correlated with both the CDR (r = −0.31, 95% CI (−0.46, −0.14), *p* < 0.001) and WMH (r = −0.17, 95% (−0.33, 0.00), *p* = 0.049). The HDL particle size correlation with the verbal memory score (r = 0.31, 95% CI (0.15, 0.45), *p* < 0.001), executive function score (r = 0.21, 95% CI (0.05, 0.34), *p* = 0.007), and CDR (r = −0.31, 95% CI (−0.45, −0.14), *p* < 0.001) remained statistically significant after adjusting for the *APOE* genotype but not the correlation with WMH (r = −0.17, 95% (−0.32, 0.02), *p* = 0.053, Table 4). Regardless of the *APOE* genotype or diagnosis status, particle size was not correlated with the CEC index (r = −0.03, 95% CI (−0.17, 0.12), *p* = 0.685) or LCAT activity (r = 0.02, 95% CI (−0.12, 0.17), *p* = 0.761).

## 4. Discussion

It is becoming increasingly apparent that disruption in the cholesterol metabolism can influence Alzheimer’s disease pathology [23], and HDL particles have a critical role in maintaining the metabolic and homeostatic processes of regulating cellular cholesterol concentrations. HDL particles in the circulation have been recognized for their potential involvement in Alzheimer’s disease, and the current evidence has linked peripheral HDL to cerebrovascular health and Alzheimer’s disease [6]. However, HDL-C alone is a poor indicator of the disease risk [24], whereas the functionality of the HDL particles may be a better indicator of disease risk, with CEC emerging as useful in identifying individuals at risk for cardiovascular disease across multiple cohorts [25,26,27]. Here, our study revealed *APOE* genotype-specific alterations in peripheral HDL CEC and LCAT activity, two critical functional metrics in the reverse cholesterol transport pathway.

Limited studies have measured HDL CEC and, to a much lesser degree, LCAT activity in Alzheimer’s disease patients. Khalil et al. [10] found impairment in the peripheral HDL CEC and LCAT activity in 39 Alzheimer’s disease patients compared to 20 healthy controls, but the *APOE* genotype was not assessed. Both Yassine et al. [28] and Marchi et al. [29] reported a reduction in cerebrospinal fluid-mediated CEC in Alzheimer’s disease patients. Such findings suggest a global decrease in the ability to efflux and remove excess cholesterol in Alzheimer’s disease. However, in our study of *APOE* genotyped controls, as well as MCI and AD patients, we did not find statistically significant alterations in peripheral HDL CEC in the MCI or AD patients without stratifying for the *APOE* genotype. To our knowledge, we report for the first time that both the CEC and LCAT activity are overall lower in *APOE3E4* carriers compared to *APOE3E3* carriers regardless of the AD diagnosis, highlighting the importance of the *APOE* genotype in the ability to efflux and transport cholesterol. *APOE3E3* MCI and AD patients exhibited lower LCAT activity, and *APOE3E3* MCI patients had significantly lower HDL CEC than the controls, while the HDL particles of *APOE3E3* AD patients exhibited a wider variability, and thus, the decrease in HDL CEC did not reach statistical significance.

On the other hand, paradoxically, *APOE3E4* AD patients had higher HDL CEC than the controls. There are three plausible explanations for this observation. First, given the fact that *APOE4* carriers are known to be at a higher risk for the earlier onset of dementia [30,31], it is possible that the *APOE3E4* carriers have higher HDL CEC as a positive compensatory response to a poorer efflux capacity over the entire lifetime. Second, *APOE3E4* AD patients had a lower prevalence of metabolic disease (diabetes or hypercholesterolemia), which is known to be linked with poorer HDL function [32,33]. Third, we observed that *APOE3E4* AD patients have more small HDL particles, and because we standardized the quantity of HDL used in the efflux experiment based on the HDL protein content, it is likely that a higher number of total HDL particles was applied to the assay compared to controls. Thus, the observed increase in HDL CEC in *APOE3E4* AD patients may simply be an observation of more total particles being able to efflux more total cholesterol rather than an actual higher cholesterol efflux capacity per particle. Most experiments measuring HDL CEC do so with apolipoprotein-B (ApoB) precipitated plasma and therefore apply the HDL dose as a percentage by volume in the cell media [25,34]. With this approach, there would also be a higher number of smaller particles applied to the assay if there were more total particles in the plasma. In our study, we isolated and purified HDL particles from plasma and applied equal amounts of HDL as the total protein. The disadvantage of measuring CEC in ApoB precipitated plasma is that this is not a direct measurement of the CEC of the HDL particles per se but of everything that remains in the plasma compartment after the ApoB-containing particles have been removed, thus the CEC measurement includes proteins and other components in plasma that could influence the ability to efflux cholesterol. The advantage of our approach is that we are measuring the CEC of the HDL particles themselves, rather than a combination of factors. One limitation of our approach for measuring the particle size by TEM is that this method does not capture all the particles in plasma and therefore does not enable quantification of the total HDL particle number. In the future, as methods become available to determine the particle count while simultaneously isolating HDL particles, the CEC assay can be dosed by equal particle number to compare the CEC on a per particle basis. Further studies are also needed to determine whether there are differences in the total particle numbers and particle size distributions in MCI and AD patients by the *APOE* genotype and to determine the HDL function and size in the other *APOE* genotypes (i.e., *APOE3E2*, *APOE4E4*, *APOE4E2*, and *APOE2E2*).

We did not find statistically significant differences in LCAT activity among the *APOE3E4* carriers. We observed overall lower LCAT activity in *APOE3E4* vs. *APOE3E3* carriers; thus, the lack of difference in LCAT activity among the *APOE3E4* carriers may be due to the fact that all individuals who are carriers of *APOE4* may already have diminished LCAT activity compared to non-*APOE4* carriers, and this reduction is not further enhanced in patients with dementia compared to the controls. Furthermore, the LCAT abundance in HDL was not measured. Thus, changes in the LCAT activity may be due to a lower abundance of LCAT protein in *APOE4* carriers, which could reflect overall lower LCAT activity.

The HDL size reflects the remodeling stage of HDL and is highly associated with the HDL composition and function. For example, it has been documented that certain HDL-associated proteins are either exclusively associated with or enriched in certain size-based HDL subclasses [13,35]. Notably, ApoE tends to associate exclusively with larger HDL particles [13]. Smaller HDL particle sizes are also associated with metabolic syndrome and chronic kidney disease [36]. We found that the HDL particle size was significantly smaller in AD and MCI patients compared to the controls overall and in the *APOE3E4* carriers but not in the *APOE3E3* carriers. Future studies are needed to better understand the relationships between the overall metabolic status and HDL particle structure and function in the context of Alzheimer’s disease and the *APOE* genotype.

Notably, we observed that HDL LCAT activity and particle size were positively correlated with the verbal memory score and negatively correlated with CDR, while the HDL particle size was additionally positively correlated with the executive function. Together, these findings suggest that disturbances in the HDL structure and function are associated with cognitive function and may be involved in Alzheimer’s disease pathology. In elderly participants, verbal memory and expression strongly predict the progression from normal cognitive function to MCI before the appearance of clinical symptoms [37]. Subtle changes in the functional measurements in participants with higher CDR scores are associated with an increased risk of converting from normal cognition to MCI [38].

Although it is not yet clear whether and how peripheral HDL particles cross the blood–brain barrier, studies have found high correlations between plasma and cerebrospinal fluid concentrations of HDL-associated apoproteins that are known not to be expressed in the CNS [39,40]. Additionally, the peripheral overexpression of human ApoA-I improved the cognitive function, reduced neuroinflammation, and protected mice from cerebral amyloid angiopathy [9]. Future studies are needed to understand the mechanisms by which peripheral HDL particles and/or their components affect the brain.

The study’s strengths include the measure of two critical functions of HDL in the reverse cholesterol transport pathway and the measurement of the HDL particle size by TEM in HDL from plasma samples obtained from a large multi-ethnic cohort of *APOE*-genotyped, well-characterized, clinically diagnosed, and/or pathologically confirmed elderly participants from the UC Davis ADRC biorepository. To our knowledge, we provide evidence for the first time of an *APOE* genotype-dependent difference in HDL functional capacity, size, and dementia and also an association between HDL functional capacity, size, and cognitive function. However, the use of cross-sectional samples limits our ability to detect changes in HDL function and size across the continuum of Alzheimer’s disease and changes in these parameters over time. Furthermore, in this study, HDL CEC was measured using the J774A.1 cell line. Future studies measuring the CEC using neuronal cells, such as neuroblastoma (SK-N-SK) cells, microglia, and other brain-relevant cell types, are needed. Another limitation is that, in this study, information on whether the participants were consuming antidiabetic agents or statins, which are commonly prescribed in AD patients and which may influence the HDL functional capacity and size, was not available, although the effects of statins on CEC remain uncertain, as reviewed elsewhere [41]. Future studies are needed to further investigate the underlying compositional and structural differences that explain the observed differences in the HDL functional capacity, particularly with regards to the *APOE* genotype. Many modifications of HDL particles, including oxidation, glycation, loss of functional components, and gain of deleterious components, have been found to influence the HDL functional capacity, even beyond the ability to efflux cholesterol [42,43,44].

## 5. Conclusions

Our findings further support earlier observations that HDL particles are implicated in Alzheimer’s disease pathology and highlight, for the first time, that there is an *APOE* genotype-dependent relationship that merits further study. Notably, our results suggest that the mechanisms of HDL deficiency in *APOE4* carriers vs. non-carriers are different, highlighting the need for further research on HDL metabolism and function in *APOE* genotyped individuals to elucidate the potential precision medicine-based approaches to improve the HDL functionality in individuals at risk for Alzheimer’s disease tailored to their *APOE* genotype. It will be important to design future studies to determine which HDL compositional and structural changes underlie the loss of function that contributes to Alzheimer’s disease pathology and how *APOE* genotype shapes these processes, so that potential therapeutic strategies to improve HDL functionality can be tested for their effectiveness in the prevention or treatment of cognitive decline.

## Figures and Tables

**Figure 1 biomedicines-10-01495-f001:**
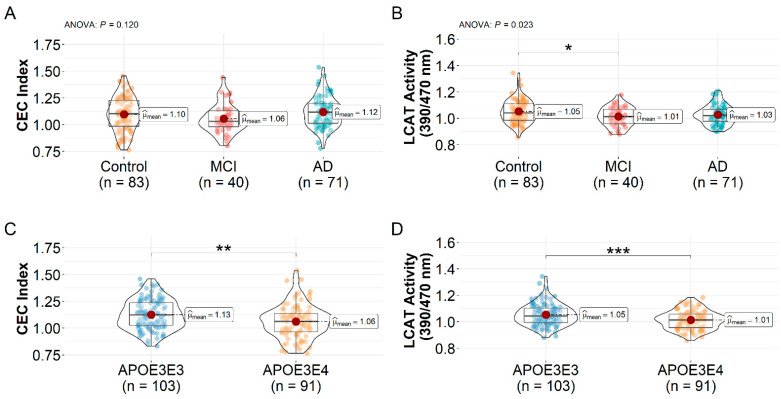
*APOE3E4* carriers have reduced HDL CEC index and LCAT activity. HDL (**A**) CEC index and (**B**) LCAT activity by diagnosis. One-way ANOVA followed by Tukey’s multiple comparison tests was used. Decrease of (**C**) CEC index and (**D**) LCAT activity in the *APOE3E4* genotype using two-sample *t*-tests. * *p* < 0.05, ** *p* < 0.01, and *** *p* < 0.001. Abbreviations: AD = Alzheimer’s disease dementia, CEC = cholesterol efflux capacity, HDL = high-density lipoproteins, LCAT = lecithin-cholesterol acyltransferase, MCI = mild cognitive impairment.

**Figure 2 biomedicines-10-01495-f002:**
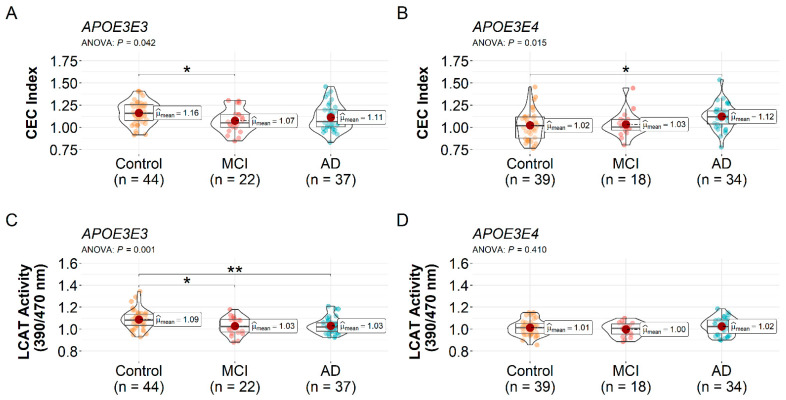
*APOE*-specific alteration in the HDL CEC index and LCAT activity amongst the control, MCI, and AD participants. The HDL CEC index within the (**A**) *APOE3E3* and (**B**) *APOE3E4* carriers by diagnosis. Decrease of LCAT activity in (**C**) *APOE3E3* patients but not (**D**) *APOE3E4* patients. One-way ANOVA followed by Tukey’s multiple comparison tests was used. * *p* < 0.05 and ** *p* < 0.01. Abbreviations: AD = Alzheimer’s disease dementia, CEC = cholesterol efflux capacity, HDL = high-density lipoproteins, LCAT = lecithin-cholesterol acyltransferase, MCI = mild cognitive impairment.

**Figure 3 biomedicines-10-01495-f003:**
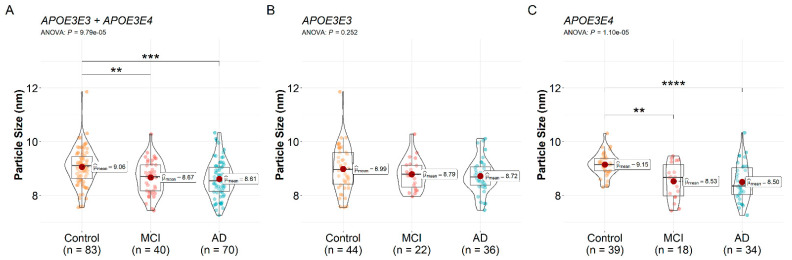
High-density lipoproteins particle size is reduced in MCI and AD patients. Mean particle diameter (nm) between (**A**) the controls, mild cognitive impairment (MCI), and Alzheimer’s disease dementia (AD) patients and controls, MCI, and AD patients stratified by the (**B**) *APOE3E3* and (**C**) *APOE3E4* genotypes. One-way ANOVA followed by Tukey’s multiple comparison tests was used. ** *p* < 0.01, *** *p* < 0.001, and **** *p* < 0.0001.

**Table 1 biomedicines-10-01495-t001:** Participant characteristics.

Characteristics	Control	MCI	AD	*F*(*df*, *df* Error) or χ^2^(*df*)	*p*-Value
*n* combined	83	40	71	n/a	n/a
*APOE3E3*	44	22	37	n/a	n/a
*APOE3E4*	39	18	34	n/a	n/a
Sex proportion, (male/female), combined	35:48	23:17	27:44	χ^2^(2) = 4.10	0.129
*APOE3E3*	18:26	12:10	15:22	χ^2^(2) = 1.34	0.511
*APOE3E4*	17:22	11:7	12:22	χ^2^(2) = 3.18	0.203
Age, years, mean ± SD, (*n*), combined	75.5 ± 7.0 (83)	78.0 ± 7.0 (40)	78.9 ± 7.2 (71)	*F*(2, 191) = 4.82	0.009 ^ab^
*APOE3E3*	75.9 ± 7.0 (44)	78.6 ± 7.7 (22)	79.7 ± 8.4 (37)	*F*(2, 100) = 2.55	0.080
*APOE3E4*	75.0 ± 7.1 (39)	77.3 ± 6.2 (18)	78.1 ± 5.7 (34)	*F*(2, 88) = 2.28	0.108
BMI, kg/m^2^, mean ± SD, (*n*), combined	28.1 ± 4.6 (81)	27.1 ± 4.2 (39)	26.6 ± 4.6 (64)	*F*(2, 181) = 2.00	0.138
*APOE3E3*	28.6 ± 4.5 (43)	27.8 ± 5.1 (21)	26.2 ± 4.7 (32)	*F*(2, 93) = 2.49	0.089
*APOE3E4*	27.5 ± 4.6 (38)	26.3 ± 2.7 (18)	27.0 ± 4.9 (32)	*F*(2, 85) = 0.42	0.658
Ethnicity proportion, (African American/Asian/Hispanic/White), combined	18:2:19:44	4:1:4:31	7:1:11:52	χ^2^(6) = 10.97	0.089
Diabetes at sample collection, %, (*n*), combined	34% (76)	7.9% (38)	4.9% (61)	χ^2^(2) = 22.94	<0.001^abc^
*APOE3E3*	39% (41)	9.5% (21)	0% (32)	χ^2^(2) = 19.30	<0.001 ^abc^
*APOE3E4*	29% (35)	5.9% (17)	10% (29)	χ^2^(2) = 5.64	0.060
History of diabetes, %, (*n*), combined	39% (83)	10% (39)	5.6% (71)	χ^2^(2) = 28.50	<0.001 ^abc^
*APOE3E3*	39% (44)	14% (21)	0% (37)	χ^2^(2) = 19.51	<0.001 ^abc^
*APOE3E4*	38% (39)	5.6% (18)	12% (34)	χ^2^(2) = 8.52	0.014 ^abc^
History of hypercholesterolemia, %, (*n*), combined	74% (81)	60% (40)	54% (70)	χ^2^(2) = 6.70	0.035 ^ab^
*APOE3E3*	62% (42)	64% (22)	47% (36)	χ^2^(2) = 2.21	0.331
*APOE3E4*	87% (39)	56% (18)	62% (34)	χ^2^(2) = 8.52	0.014 ^abc^
History of hypertension, %, (*n*), combined	72% (83)	65% (40)	76% (71)	χ^2^(2) = 1.56	0.459
*APOE3E3*	75% (44)	77% (22)	76% (37)	χ^2^(2) = 0.04	0.980
*APOE3E4*	69% (39)	50% (18)	76% (34)	χ^2^(2) = 3.84	0.147

Abbreviations: AD = Alzheimer’s disease dementia, BMI = body mass index, MCI = mild cognitive impairment. ANOVA were performed for continuous variables (age and BMI), and chi-squared (χ^2^) were performed for categorical variables (sex proportion, ethnicity proportion, diabetes at sample collection, history of diabetes, history of hypercholesterolemia, and history of hypertension). If significance is reached by ANOVA or χ^2^, Tukey’s or Bonferroni’s post hoc comparison was carried out, respectively. ^a^ Significance across all three groups. ^b^ Significance between the control compared with AD. ^c^ Significance between the control compared with MCI.

**Table 2 biomedicines-10-01495-t002:** Comparison of the HDL CEC index, LCAT activity, and particle size across diagnoses.

Genotype	Group			ANOVA				Post hoc Comparison *p*-Value		
	Control	MCI	AD	*F*	*df*	*df* Error	*p*-Value	Control vs. MCI	Control vs. AD	MCI vs. AD
CEC Index										
*APOE3E3* + *APOE3E4*	1.10 ± 0.16	1.06 ± 0.14	1.12 ± 0.14	2.14	2	191	0.120	-	-	-
*APOE3E3*	1.16 ± 0.13	1.07 ± 0.13	1.11 ± 0.15	3.26	2	100	0.042	0.042	0.246	0.551
*APOE3E4*	1.02 ± 0.16	1.03 ± 0.14	1.12 ± 0.14	4.38	2	88	0.015	0.964	0.016	0.114
LCAT Activity (390/470 nm)										
*APOE3E3* + *APOE3E4*	1.05 ± 0.09	1.01 ± 0.07	1.03 ± 0.07	3.87	2	191	0.023	0.030	0.118	0.661
*APOE3E3*	1.09 ± 0.09	1.03 ± 0.08	1.03 ± 0.07	7.00	2	100	0.001	0.012	0.004	0.995
*APOE3E4*	1.01 ± 0.07	1.00 ± 0.06	1.02 ± 0.07	0.91	2	88	0.410	-	-	-
HDL Particle Diameter (nm)										
*APOE3E3* + *APOE3E4*	9.06 ± 0.69	8.67 ± 0.63	8.61 ± 0.68	9.70	2	190	<0.001	0.008	<0.001	0.901
*APOE3E3*	8.99 ± 0.85	8.79 ± 0.58	8.72 ± 0.64	1.40	2	99	0.252	-	-	-
*APOE3E4*	9.15 ± 0.42	8.53 ± 0.67	8.50 ± 0.70	13.04	2	88	<0.001	0.001	<0.001	0.985

ANOVA were performed to compare group differences. If the significance is reached by ANOVA, Tukey’s post hoc comparison was carried out. Values are represented as the mean ± standard deviation. Abbreviations: AD = Alzheimer’s disease dementia, CEC = cholesterol efflux capacity, HDL = high-density lipoproteins, LCAT = lecithin-cholesterol acyltransferase, MCI = mild cognitive impairment.

**Table 3 biomedicines-10-01495-t003:** Cognitive, functional, and magnetic resonance imaging scores across the diagnoses.

Variable		Group			Kruskal–Wallis Test		
		Control *n* = 83	MCI *n* = 40	AD *n* = 71	χ2	*df*	*p*-Value
Cognitive	Verbal memory score,	0.24	−0.97	−1.41	87.72	2	<0.001
	median (25th, 75th),	(−0.39, 0.67)	(−1.37, −0.73)	(−1.90, −1.06)			
	(*n*)	(69)	(35)	(40)			
	Executive function score,	−0.03	−0.37	−0.96	54.29	2	<0.001
	median (25th, 75th),	(−0.27, 0.44)	(−0.60, −0.07)	(−1.52, −0.55)			
	(*n*)	(68)	(35)	(44)			
	Semantic memory score,	0.50	0.23	−0.50	32.95	2	<0.001
	median (25th, 75th),	(−0.08, 0.91)	(−0.12, 0.66)	(−1.01, 0.17)			
	(*n*)	(68)	(35)	(43)			
	Spatial score,	0.42	0.14	−0.86	32.09	2	<0.001
	median (25th, 75th),	(−0.18, 0.83)	(−0.21, 0.48)	(−1.36, −0.05)			
	(*n*)	(67)	(35)	(37)			
Functional	CDR sum of boxes,	0.00	3.00	5.00	94.79	2	<0.001
	median (25th, 75th),	(0.00, 0.50)	(1.50, 3.50)	(3.38, 7.00)			
	(*n*)	(59)	(33)	(40)			
Imaging ^a^	WMH volume,	0.00	0.01	0.01	18.27	2	<0.001
	median (25th, 75th),	(0.00, 0.01)	(0.01, 0.02)	(0.00, 0.01)			
	(*n*)	(65)	(33)	(40)			

^a^ White matter hyperintensities (WMH) volume was normalized to total intracranial volume. Abbreviations: AD = Alzheimer’s disease dementia, CDR = clinical dementia rating, MCI = mild cognitive impairment.

**Table 4 biomedicines-10-01495-t004:** Correlation analysis across diagnoses adjusting for *APOE* genotype between HDL function and size with either cognitive, functional, and magnetic resonance imaging scores.

	Cognitive				Functional	Imaging ^a^
Characteristic	Verbal Memory Score	Executive Function Score	Semantic Memory Score	Spatial Score	CDR Sum of Boxes	WMH Volume
*n*	144	147	146	139	132	138
CEC Index						
r	0.02	0.01	−0.10	−0.01	0.04	0.12
95% CI	(−0.15, 0.19)	(−0.15, 0.18)	(−0.26, 0.07)	(−0.27, 0.07)	(−0.14, 0.21)	(−0.05, 0.29)
*p*-value	0.826	0.877	0.234	0.245	0.693	0.149
Adjusted ^b^ r	0.02	0.03	−0.07	−0.08	0.05	0.12
Adjusted ^b^ 95% CI	(−0.14, 0.19)	(−0.15, 0.19)	(−0.23, 0.09)	(−0.24, 0.08)	(−0.12, 0.23)	(−0.04, 0.29)
Adjusted ^b^ *p*-value	0.824	0.720	0.397	0.334	0.571	0.160
LCAT Activity (390/470 nm)						
r	0.17	0.13	−0.13	−0.10	−0.20	−0.05
95% CI	(0.01, 0.33)	(−0.03, 0.29)	(−0.29, 0.04)	(−0.26, 0.08)	(−0.37, −0.03)	(−0.22, 0.12)
*p*-value	0.037	0.105	0.132	0.265	0.019	0.562
Adjusted ^b^ r	0.18	0.15	−0.10	−0.08	−0.20	−0.06
Adjusted ^b^ 95% CI	(0.01, 0.33)	(−0.02, 0.29)	(−0.27, 0.07)	(−0.26, 0.07)	(−0.35, −0.02)	(−0.21, 0.10)
Adjusted ^b^ *p*-value	0.033	0.069	0.231	0.358	0.025	0.512
Particle Size (nm)						
r	0.31	0.21	0.07	0.08	−0.31	−0.17
95% CI	(0.15, 0.45)	(0.04, 0.36)	(−0.10, 0.24)	(−0.09, 0.25)	(−0.46, −0.14)	(−0.33, 0.00)
*p*-value	<0.001	0.013	0.396	0.330	<0.001	0.049
Adjusted ^b^ r	0.31	0.21	0.08	0.09	−0.31	−0.17
Adjusted ^b^ 95% CI	(0.15, 0.45)	(0.05, 0.34)	(−0.08, 0.23)	(−0.08, 0.25)	(−0.45, −0.14)	(−0.32, 0.02)
Adjusted ^b^ *p*-value	<0.001	0.007	0.331	0.313	<0.001	0.053

Spearman’s correlation coefficients (r) of the CEC index and LCAT activity with cognitive and magnetic resonance imaging assessments across the diagnoses (control, MCI, and AD). AD = Alzheimer’s disease dementia, CDR = clinical dementia rating, CEC = cholesterol efflux capacity, HDL = high-density lipoproteins, LCAT = lecithin-cholesterol acyltransferase, MCI = mild cognitive impairment. ^a^ White matter hyperintensities (WMH) volume was normalized to the total intracranial volume. ^b^ Adjusted for the *APOE* genotype.

## Data Availability

The data used in the study are available from the corresponding author upon request.

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
