# Peer review of "High-Density Lipoprotein Changes in Alzheimer’s Disease Are *APOE* Genotype-Specific"

_biomedicines, 2022, doi:10.3390/biomedicines10071495_

Round 1

Reviewer 1 Report

The paper entitled “High-density lipoprotein changes in Alzheimer’s disease are 2 APOE genotype-specific” is very well structured and presented. The main topics of the paper are well introduced, and the methodology and results are well organized. This work showed the impact of APOE3E4 genotype in HDL function and size in AD patients.

I only suggest that the authors clarify in the conclusion that the method used does not enable the HDL particle number quantification, which is a relevant information.

Reviewer 2 Report

REVEIWER COMMENTS

Hong et al.

In this manuscript, the authors have studied cholesterol efflux capacity (CEC), lecithin-cholesterol acyltransferase (LCAT) activity, and HDL particle diameter in plasma derived HDL particles from MCI and AD patients and a control cohort in relation to ApoE genotype. The authors have used standard biochemistry and molecular imaging, combined with statistics and correlation to clinical parameters. The authors conclude that both CEC and LCAT activity are overall lower in APOE3E4 carriers compared to APOE3E3 carriers regardless of AD diagnosis. The authors also assert that the changes in CEC, LCAT activity and HDL particle size are correlated to some of the cognitive performance tests. Overall, the study is well designed and the topic is also relevant to AD pathogenesis. The manuscript is well written and contains sufficient discussion on the strengths and weaknesses of the study and conclusion drawn from the available data.

I only have a few minor suggestions that the author should either address experimentally or discuss as potential limitations:

           CEC assays: The authors have used only one cell line J774A.1 (monocytes/macrophages). Ideally, the findings should be replicated in a different cell type, and preferably in a neuronal cell line such as SK-N-SH or similar. See for example  DOI: 10.1042/BCJ20180068 

           LCAT assays: Here a commercial kit is used. However there is no biochemical indication if the changes in the LCAT activity are due to the degree of LCAT abundance in the purified HDL pool. The authors should at least comment on such possibility. 

           HDL size. The authors describe the TEM procedure and image analyses in the Methods section. It would be useful for the readers to prepare a short schematic how the image analyses were done and include a few representative images. 

           With regards to the population characteristics (control, MCI and AD), have the authors considered whether the participants were being treated with anti-diabetic or lipid-lowering agents. How does that affect the distribution of measures in Fig1-3. 

           The authors should also highlight the salient differences in plasma HDL and HDL in the CNS. This is important since the authors have measured the features in plasma HDL and should discuss the limitations with regards to interpretations of cognitive tests. 

           Another minor comment: The updated disease nomenclature for AD is ‘Alzheimer disease’ without the apostrophe....see for example Escourolle & Poirier or Stedman’s Medical Acronyms
